# Arterial stiffness and cardiac dysfunction in Hutchinson–Gilford Progeria Syndrome corrected by inhibition of lysyl oxidase

Ryan von Kleeck[1,2] , Emilia Roberts[1,3], Paola Castagnino[1,3], Kyle Bruun[1], Sonja A Brankovic[1,2], Elizabeth A Hawthorne[1], Tina Xu[1], John W Tobias[4], Richard K Assoian[1,2,3]

Arterial stiffening and cardiac dysfunction are hallmarks of premature aging in Hutchinson–Gilford Progeria Syndrome (HGPS), but the molecular regulators remain unknown. Here, we show that the LaminA[G609G] mouse model of HGPS recapitulates the premature arterial stiffening and early diastolic dysfunction seen in human HGPS. Lysyl oxidase (LOX) is up-regulated in the arteries of these mice, and treatment with the LOX inhibitor, $\beta$-aminopropionitrile, improves arterial mechanics and cardiac function. Genome-wide and mechanistic analysis revealed reduced expression of the LOX-regulator, miR-145, in HGPS arteries, and forced expression of miR-145 restores normal LOX gene expression in HGPS smooth muscle cells. LOX abundance is also increased in the carotid arteries of aged wild-type mice, but its spatial expression differs from HGPS and its up-regulation is independent of changes in miR-145 abundance. Our results show that miR-145 is selectively misregulated in HGPS and that the consequent up-regulation of LOX is causal for premature arterial stiffening and cardiac dysfunction.

## Introduction

Hutchinson–Gilford Progeria Syndrome (HGPS) is a rare genetic disease of premature aging. HGPS is caused by an autosomal dominant mutation in *LMNA*, the gene encoding Lamin A (Capell et al, 2007; Gonzalo et al, 2016). The mutation is "silent" (LaminA[G608G]) but leads to a defect in biosynthetic processing of the Lamin A precursor and results in a truncated, farnesylated version of Lamin A called progerin, which is defective in its localization within the nucleus (Capell et al, 2007; Gonzalo et al, 2016). Children with HGPS typically die in their teenage years as a consequence of cardiovascular disease (atherosclerosis, myocardial infarction, heart failure, and/or stroke) (Capell et al, 2007; Gonzalo et al, 2016). These cardiovascular consequences occur in the

absence of high cholesterol or triglycerides (Gordon et al, 2005), suggesting that cholesterol-independent risk factors must be at play.

Intriguingly, the arteries of HGPS patients are abnormally stiff (Gerhard-Herman et al, 2011), and arterial stiffness is a cholesterol-independent risk factor for cardiovascular disease (Mitchell et al, 2010). Increased arterial stiffness is also a hallmark of normal aging and increases substantially after the age of 50 in healthy males and females (Mitchell et al, 2007). However, the arterial stiffness in HGPS children resembles that of ~60 yr-old individuals (Gerhard-Herman et al, 2011), indicating a striking acceleration of arterial stiffening. Pathologically stiff arteries increase load on the heart (Safar, 2010; Laurent & Boutouyrie, 2015), which can then have systemic consequences and correlates with cardiac abnormalities such as left ventricular hypertrophy and diastolic dysfunction (Mottram et al, 2005; Kim et al, 2017). Consistent with these relationships, left ventricular hypertrophy and, particularly, diastolic dysfunction are also observed in HGPS children (Prakash et al, 2018).

The composition of the ECM plays a critical role in arterial stiffness (Cox & Erler, 2011). The stiffness of the major arteries relies on a balance of elastin and collagen fibers (Humphrey et al, 2014), which maintains a proper response to blood pressure. Elastin allows for recoil at lower load and dampens cyclic loading from the beating heart, whereas fibrillar collagens contribute to the strain-stiffening property of arteries at high load (Kohn et al, 2015). Arteries express three main fibrillar collagens, with collagen-I > collagen-III > collagen-V in abundance (Prockop & Kivirikko, 1995; Hulmes, 2008). Increased collagen deposition as well as elastin fragmentation has been observed in the arteries of HGPS children at autopsy (Merideth et al, 2008; Olive et al, 2010) and in aged arteries (Tsamis et al, 2013; Kohn et al, 2015).

In addition to increases in fibrillar collagens, the mechanical properties of tissues are regulated by matrix remodeling enzymes (Freitas-Rodríguez et al, 2017; Lampi & Reinhart-King, 2018), especially lysyl oxidase (LOX) and its family members (LOXL1, LOXL2, LOXL3, and LOXL4) (Yamauchi & Sricholpech, 2012; Nilsson et al,

[1]Department of Systems Pharmacology and Translational Therapeutics, University of Pennsylvania, Philadelphia, PA, USA    [2]Center for Engineering MechanoBiology, University of Pennsylvania, Philadelphia, PA, USA    [3]Institute of Translational Medicine and Therapeutics and University of Pennsylvania, Philadelphia, PA, USA    [4]Penn Genomic Analysis Core and University of Pennsylvania, Philadelphia, PA, USA

Correspondence: assoian@pennmedicine.upenn.edu

2016; Steppan et al, 2019). These enzymes covalently cross-link newly synthesized collagen and elastin fibers to enhance their stability and tensile strength (Baker et al, 2013; Tsamis et al, 2013; Herchenhan et al, 2015), and LOX overexpression is commonly seen in stiffness-related pathologies (Kagan, 1994; Desmoulière et al, 1997). However, because elastin biosynthesis ends early in life (Davidson et al, 1986; Kelleher et al, 2004; Mithieux & Weiss, 2005; Wagenseil & Mecham, 2012; Tsamis et al, 2013), LOX-mediated cross-linking is thought to target newly synthesized collagens in normal aging and promote arterial stiffening.

Because HGPS is a very rare genetic disease (Olive et al, 2010; Gerhard-Herman et al, 2011), the number of HGPS patients is very small, and the detailed analysis of early and potentially causal events in the pathogenesis of HGPS must rely heavily on animal models. Osorio et al (2011) developed a knock-in mouse (hereafter called HGPS mouse) that models the human disease by expressing murine progerin (Lmna^{G609G/G609G}) at the endogenous *Lmna* locus. These mice display many traits of the human disease including premature death. Arterial stiffening, left ventricular diastolic dysfunction, and an increase in the expression of several collagens, including collagen-III, IV, and V is also apparent in these mice near the end of their lifespan (del Campo et al, 2019; Murtada et al, 2020). Whether these changes are primary or secondary effects of progerin expression is not known.

In this report, we have examined arterial mechanics, cardiac function, and molecular mechanism in young HGPS mice (2 mo of age, before the onset of morbidity) in an effort to identify early and potentially causal events in disease progression. We report that arterial stiffening in HGPS is strikingly premature and associated with diastolic dysfunction. Arterial LOX is strongly up-regulated in these young mice, and systemic LOX inhibition improves arterial mechanics and cardiac function. Genome-wide profiling and cell-based mechanistic analysis indicates that a pathologic misregulation of miR-145 in HGPS is a critical and selective determinant of elevated LOX expression.

# Results

### Premature arterial stiffening in HGPS mice

We began our studies of arterial stiffening in HGPS by comparing mechanics and ECM remodeling in the arteries of WT mice versus the progerin-expressing HGPS mouse. We focused on the carotid artery, a prominent site of atherosclerotic lesion formation, as its occlusion is thought to be responsible for induction of stroke in HGPS children (Gerhard-Herman et al, 2011). Freshly isolated carotid arteries were stretched below, at, and above their in vivo lengths at constant pressure to provide insight into axial arterial stress (Fig 1A1, left). The artery was also stretched to its in vivo length (referred to as the In Vivo Stretch; IVS), and then pressurized incrementally to provide insight into circumferential vessel mechanics (Fig 1A2, right). Changes in outer diameter, wall thickness, and inner radius were used to generate axial and circumferential stress–stretch relationships, in which axial and circumferential stiffness is represented as the stress the artery experiences for a

defined increase in stretch. These stress–stretch data were also used to derive the axial and circumferential tangent modulus (versus stretch), an additional way to estimate arterial stiffness.

We recently used these analyses to examine the mechanical and geometric properties of carotid arteries of C57BL/6 mice aged from 2 to 24 mo; the results showed decreases in IVS as well as increased axial and circumferential arterial stiffness with age, beginning at 12 mo and becoming even more evident by 24 mo (Brankovic et al, 2019). Therefore, we also compared young (2-mo) HGPS mice to 24-mo WT mice to understand how arterial mechanics are affected early in HGPS and how they compare with the changes that occur in normal aging. In addition, our use of the progerin-expressing HGPS mouse allowed us to analyze sex differences in arterial biomechanics that have not been attainable given the small numbers of HGPS girls and boys.

Changes in outer diameter, inner radius, and wall thickness with pressure (Fig S1A–F) were measured and used to establish stress–stretch relationships in carotid arteries from 2-mo WT and HPGS mice. The circumferential stress–stretch (Fig 1B and D) and tangent modulus (Fig S2B and D) curves of young HGPS and aged WT mice were left-shifted for both sexes indicating increased circumferential stiffness. Axially, the IVS was reduced in HGPS male and female mice, but the magnitude was less than that seen in old WT mice of either sex (Table 1). The axial stress–stretch (Fig 1C and E) and tangent modulus (Fig S2A and C) curves suggested that axial stiffening can occur early in HGPS but may be less prominent than circumferential stiffening, particularly in the males. Collectively, these data indicate that arterial stiffening in HGPS mice is premature as it is in the human syndrome, but the early effect is anisotropic, especially in males, an insight that has not been obtainable by pulse-wave velocity studies of HGPS children (Gerhard-Herman et al, 2011).

Although reduced smooth muscle cell (SMC) number has been reported in the ascending aorta of HGPS children at autopsy (Hamczyk & Andrés, 2019), histological analysis of 2-mo WT and HGPS carotid arteries showed similar cellularity (Fig 2A) and numbers of SMC nuclei in the medial layer of young mouse carotids (Fig 2B). We observed a small increase in the abundance of the senescence marker p16^{INK4A} in the carotid medial layer of 2-mo HGPS mice (Fig 2C and D), but other markers of late HGPS vascular lesions including increases in calcium content (Merideth et al, 2008; Olive et al, 2010), apoptotic cells, and changes in elastin integrity were not seen in the HGPS carotid arteries at this early time-point (Figs 2E and F and S3A–C). Collectively, these results indicate that arterial stiffening is a very early event in the progression of HGPS and that our experimental conditions have the potential to identify initiating events before the onset of gross HGPS arterial pathology.

### Canonical up-regulation of fibrillar collagens is lacking in early HGPS

Tissue remodeling and stiffening often involves increased amounts of fibrillar collagens and/or elastin fragmentation. Collagen-I is the major strain-stiffening component of the arterial ECM, but an immunostaining analysis of carotid sections was unable to detect statistically significant increases in collagen-I in either the medial or adventitial layers of 2-mo HGPS carotid arteries as compared

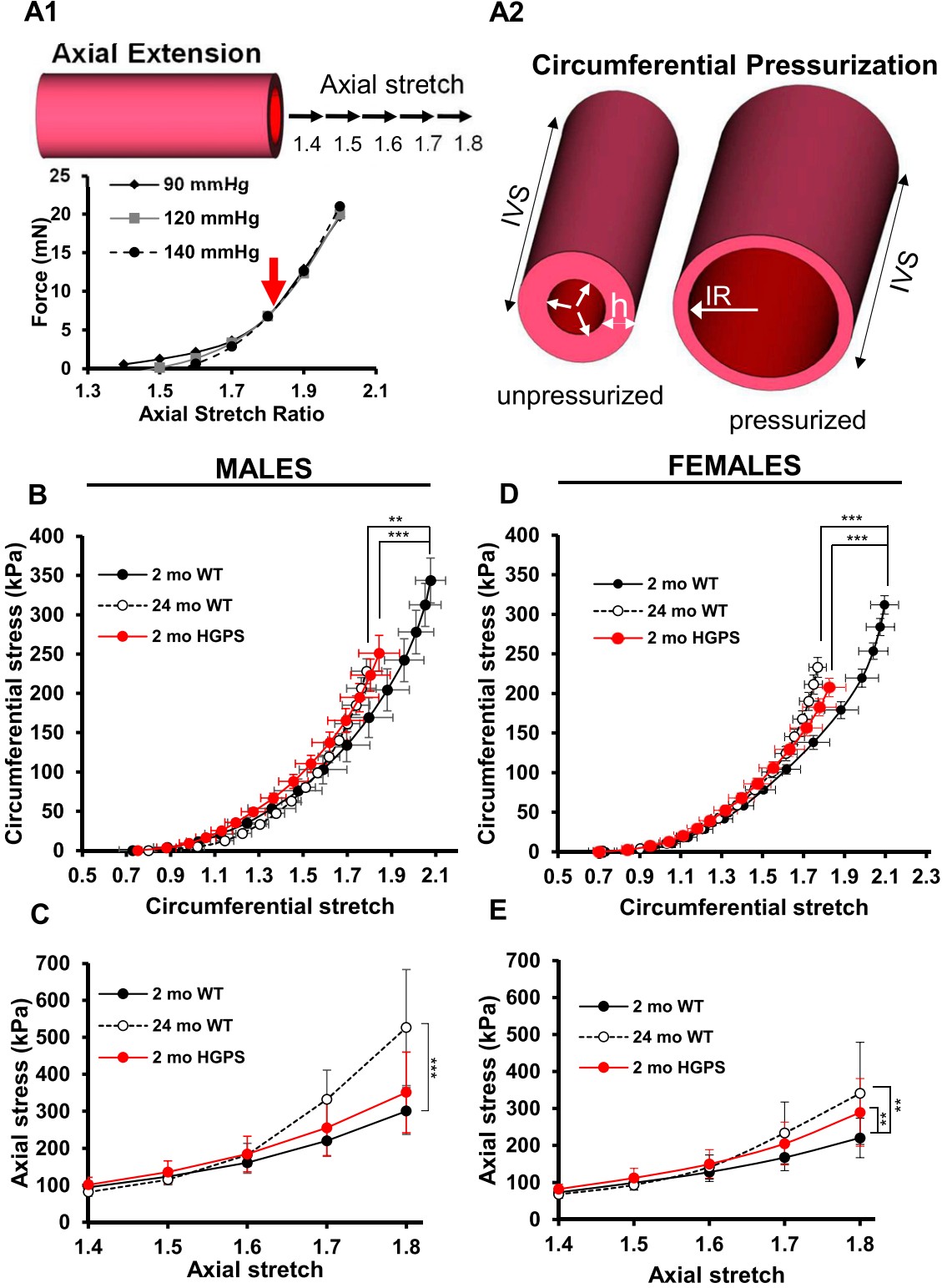

**Figure 1.   Mechanical properties of isolated carotid arteries display premature circumferential stiffening in Hutchinson–Gilford Progeria Syndrome (HGPS) male and female mice.**
**(B, C, D, E)** Carotid arteries from male (panel B, C) and female (panel D, E) 2-mo (n = 7 per sex) and 24-mo (n = 5 per sex) WT and 2-mo Hutchinson–Gilford Progeria Syndrome (n = 6 per sex) mice were analyzed by pressure myography. **(A1)** Arteries are axially stretched at multiple pressures, and the intersection point of the stretch–force curves defines the "IVS." The graph displays a representative axial stretch–force curve for a 2-mo WT mouse carotid artery (red arrow shows the IVS). **(A2)** Arteries are brought to their individual IVS and pressurized. White arrows indicate circumferential deformation with pressure (IR, inner radius; h, wall thickness).

**Table 1. In Vivo Stretch (IVS) values for WT and Hutchinson–Gilford Progeria Syndrome (HGPS) mice.**

| Genotype | Age | Sex | IVS (mean) | IVS (SD) | n | P (relative to 2-mo WT) |
|---|---|---|---|---|---|---|
| WT | 2 mo | M | 1.85 | 0.04 | 13 | Reference |
| WT | 24 mo | M | 1.72 | 0.05 | 5 | *** |
| HGPS | 2 mo | M | 1.79 | 0.10 | 9 | NS |
| WT | 2 mo | F | 1.88 | 0.07 | 7 | Reference |
| WT | 24 mo | F | 1.74 | 0.04 | 5 | * |
| HGPS | 2 mo | F | 1.79 | 0.07 | 10 | * |

Significance was determined by Mann–Whitney test in comparison to the IVS of 2-mo WT mice. NS; not significant.

with age-matched WT controls (Figs 3A and B and S4A and B). Of the minor fibrillar collagens, these young HGPS carotid arteries showed a statistically significant but small increase in medial collagen-III and no change in the abundance of collagen-V (Figs 3A and B and S4). Similar to observations in humans (Greenberg, 1986), Transmission Electron Microscopy (TEM) revealed that collagen in the medial layer of both WT and HGPS mouse carotid arteries was restricted to the elastin folds (Fig S5A). Whereas the area of collagen within these folds was significantly greater in HGPS, the magnitude of the effect was small (Fig S5B and C).

### Up-regulation of lysyl oxidase is an early event in HGPS and causal for premature arterial stiffening

In contrast to the results with collagens, immunostaining readily detected a pronounced increase in abundance of LOX in the medial layer of carotid arteries from 2-mo HGPS mice as compared with age-matched WT mice (Fig 3C and D). To determine the onset of this LOX up-regulation in HGPS, we examined LOX abundance in 1-mo mice. Although medial LOX levels were slightly increased in 1-mo HGPS mice, the effect was not significant and much less pronounced than at 2 mo (Fig 3C and D). Adventitial LOX abundance was not affected by age or genotype (Fig 3C and D).

To determine if the minimal increase in abundance of LOX in 1-mo HGPS mice correlated with changes in arterial stiffening, we used pressure-outer diameter, inner radius, and wall thickness measurements (Fig S6A–C) to generate circumferential and axial stress–stretch (Fig S6D and E) and tangent modulus (Fig S6F and G) curves for these very young WT and HGPS mice. The pronounced genotype differences in the circumferential stress–stretch relationship (compare Figs 1 and S6E) and tangent modulus curves (Fig S6G) were much attenuated at 1 mo. Axial arterial mechanics in 1-mo HGPS carotid arteries, as judged by the IVS (Table S1) and stress–stretch and tangent modulus curves (Fig S6D and F), were similar to the WT controls. Thus, the molecular analysis (Fig 3C and D) and functional testing (Figs 1 and S6) are in agreement and indicate that the time-dependent increase in medial LOX abundance in HGPS mice begins between 1 and 2 mo of age and is coincident with the onset of acute arterial stiffening.

We treated HGPS mice with the highly selective LOX enzymatic inhibitor, β-aminopropionitrile (BAPN) (Tang et al, 1983; Jung et al, 2003; Kim et al, 2003; Lee & Kim, 2006; Rodriguez et al, 2010; Finney et al, 2014), to determine the role of lysyl oxidase activity in the early arterial stiffening of HGPS. We administered BAPN between 1- and 2 mo of age, when the levels of medial LOX were increasing (Fig 3C and D) and used a BAPN concentration and injection regime that we previously showed to be effective in altering fibrillar collagen structure and reducing arterial stiffness in vivo (Kothapalli et al, 2012). The area of collagen within elastin folds of carotid arteries was similar with or without BAPN treatment as was the expression of arterial LOX (Fig S7A–D).

Biaxial inflation-extension tests examined the effect of LOX inhibition on arterial mechanics of the HGPS carotid arteries. Pressure-outer diameter, inner radius and wall thickness measurements (Fig S8A–C) were used to generate circumferential and axial stress–stretch curves (Fig 4A and B), and these (together with the corresponding tangent modulus graphs; Fig S8D and E) showed that BAPN reduced circumferential stiffness of the HGPS carotid arteries without comparable effect on the WT controls (Figs 4A and B and S8D). In fact, the circumferential stretch of BAPN-treated HGPS mice showed no statistical difference from those of WT mice treated with PBS (Figs 4A and S8D). The IVS was not affected by BAPN (Table 2), but curiously, the axial stress–stretch curves showed small but opposing effects in HGPS and WT mice (Figs 4B and S8E). Collectively, the elevated expression of LOX coincident with premature circumferential arterial stiffening and the correction of this defect by BAPN indicates that early LOX induction plays an important causal role in the premature circumferential arterial stiffening of HPGS.

In addition to arterial stiffness, HGPS patients display cardiac abnormalities, particularly diastolic dysfunction, which may increase the risk for death (Halley et al, 2011; Prakash et al, 2018). As diastolic dysfunction has been positively correlated with arterial stiffness (Mottram et al, 2005; Kim et al, 2017), we asked if the BAPN-mediated reduction in circumferential arterial stiffness of HGPS mice was also associated with improved diastolic function. HGPS mice were treated with vehicle or BAPN using conditions shown in Fig 4A and B. Echocardiography was performed to measure E (mitral

**(B, D)** Circumferential stretch–stress curves for male and female mice, respectively. Each data point corresponds to the circumferential stress and stretch at increments of 10 mm Hg starting from 0 to 140 mm Hg. **(C, E)** Axial stretch–stress curves, determined at 90 mm Hg, for male and female mice, respectively. Results in panels (B) and (D) show means ± SE and results in panel (C) and (E) show means ± SD. Statistical significance in all panels was determined by two-way ANOVA in comparison with 2-mo WT mice.

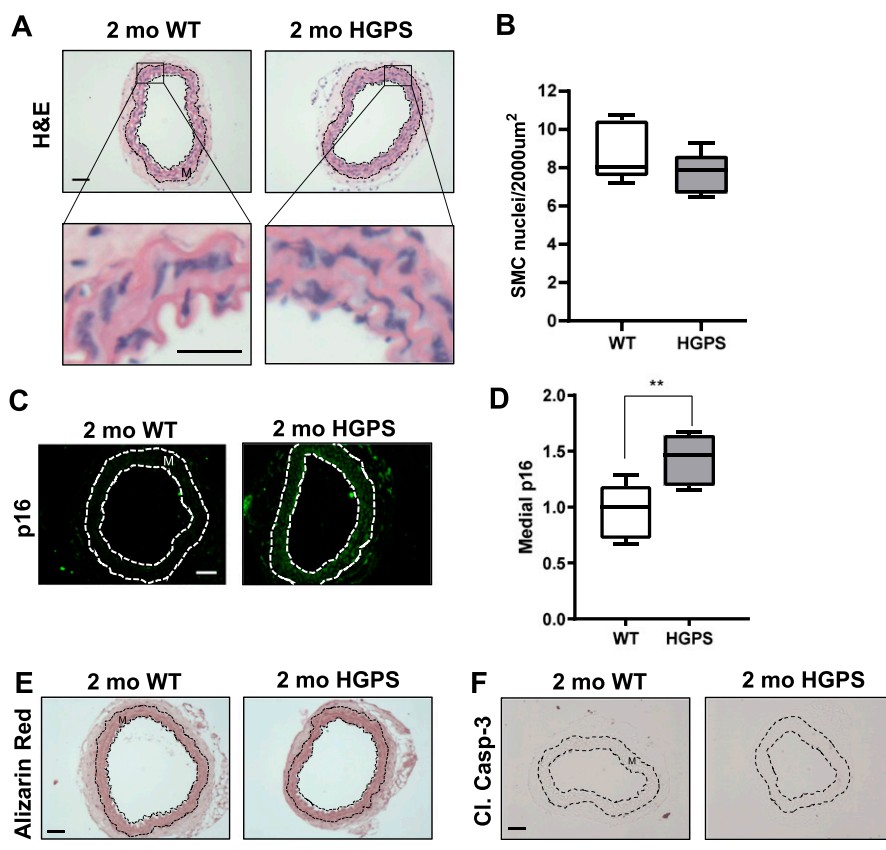

**Figure 2. Carotid arteries from 2-mo WT and Hutchinson–Gilford Progeria Syndrome (HGPS) mice display similar histology.**
**(A, B)** Carotid artery cross sections from 2-mo WT and Hutchinson–Gilford Progeria Syndrome mice were stained with H&E (scale bar = 50 $\mu$m and inset bar = 25 $\mu$m), and the number of medial smooth muscle cell nuclei was quantified from H&E images (n = 7–10 carotids per genotype with three sections analyzed per mouse). **(C, D)** Carotid cross sections were immunostained for the senescence marker, p16INK4A, and the level of medial p16INK4A was normalized to the mean signal intensity of carotid sections from 2-mo WT mice; (n = 6–10 per genotype). Scale bar = 50 $\mu$m. Statistical significance in (B) and (D) was determined by Mann–Whitney tests. **(E, F)** Carotid cross sections were stained with Alizarin Red (n = 5 per genotype) and (F) immunostained for cleaved caspase-3 (n = 5 per genotype). Scale bars in (E) and (F) = 50 $\mu$m. The arterial media (M) is outlined with dashed lines.

peak velocity of early filling) and E' (early diastolic mitral annular velocity) (Fig S9A and B). We then determined (E/E′) as a measure of diastolic dysfunction. E/E′ was improved in the BAPN-treated HGPS mice and similar to that of age- and sex-matched WT mice (Fig 4C). Although less common, left ventricular hypertrophy has been detected in HGPS children, mostly as they age (Prakash et al, 2018). Consistent with these studies, the left ventricular mass index, a marker of LV hypertrophy, trended lower in the BAPN-treated HGPS mice (Fig 4D). Systolic, diastolic, and pulse pressures were not affected by BAPN (Fig S9C–E).

To determine if the morphologic and molecular characteristics of the HGPS carotid extended to other large arteries, we repeated the analyses shown in Figs 2 and 3 in 2-mo WT and HGPS aortas. Indeed, results in the aortas were similar to those seen in the carotid arteries with regard to overall morphology, SMC number, p16 expression, calcification, and apoptosis (Fig S10A–F). We also found that these 2-mo WT and HGPS mice had similar expression levels of collagen-I, collagen-III, and collagen-V protein (Fig S11A and B) and mRNAs (Fig S11C and D) in the aortic media and adventitia, results that again resembled those seen in the carotid. The only exceptions were the modest increases in p16 and collagen-III, which were statistically significant in the HGPS carotid media but not in the aortic media (Figs S10C and S11B, respectively). Importantly, the notably increased medial abundance of LOX in the HGPS carotid artery was also seen in the HGPS aorta (Fig 5A–D). In addition, the relatively larger aortic mass allowed us to demonstrate that the increased abundance of arterial LOX protein in HGPS was accompanied by increased LOX enzymatic activity (Fig 5E).

## Reduced expression of miR-145 underlies the misregulation of arterial LOX in HGPS SMCs

We used isolated aortas and primary aortic SMCs from age-matched WT and HGPS mice to explore the molecular mechanism responsible for up-regulation of arterial LOX. We first found that the increased abundance of LOX protein we observed in the arterial media of 2-mo HGPS mice was accompanied by a preferential increase in medial rather than adventitial LOX mRNA (Fig 5F and G). As in the carotid, this up-regulation was minimally detected in younger (1-mo) HPGS mice (Fig S12A). We then extended the mRNA analysis to all of the LOX family members: LOXL4 mRNA was increased significantly in 2-mo HGPS aortic media and adventitia (Fig 5F and G), but a comparison of Δ-Ct values revealed that arterial expression of LOXL4 is very low relative to the LOX isoform (Fig S12B). Continued mechanistic analysis therefore focused on regulation of the LOX isoform in vascular SMCs.

To identify molecular mechanism(s) upstream of LOX gene expression in HGPS, we performed a genome-wide analysis of descending aortas from 2-mo WT and HGPS mice in addition to 24-mo WT mice. A principal component analysis showed clustering amongst the replicates and separation between the experimental conditions (Fig S13A). Using a 1.5× fold change and an adjusted *P*-value of <0.001 as cut-offs, we identified nearly 4,000 differentially expressed genes (DEGs) between 2-mo WT and HGPS aortas, and a gene ontology analysis identified "ECM" among the most DEG categories (Fig S13B). We then used Ingenuity Pathway Analysis (IPA)

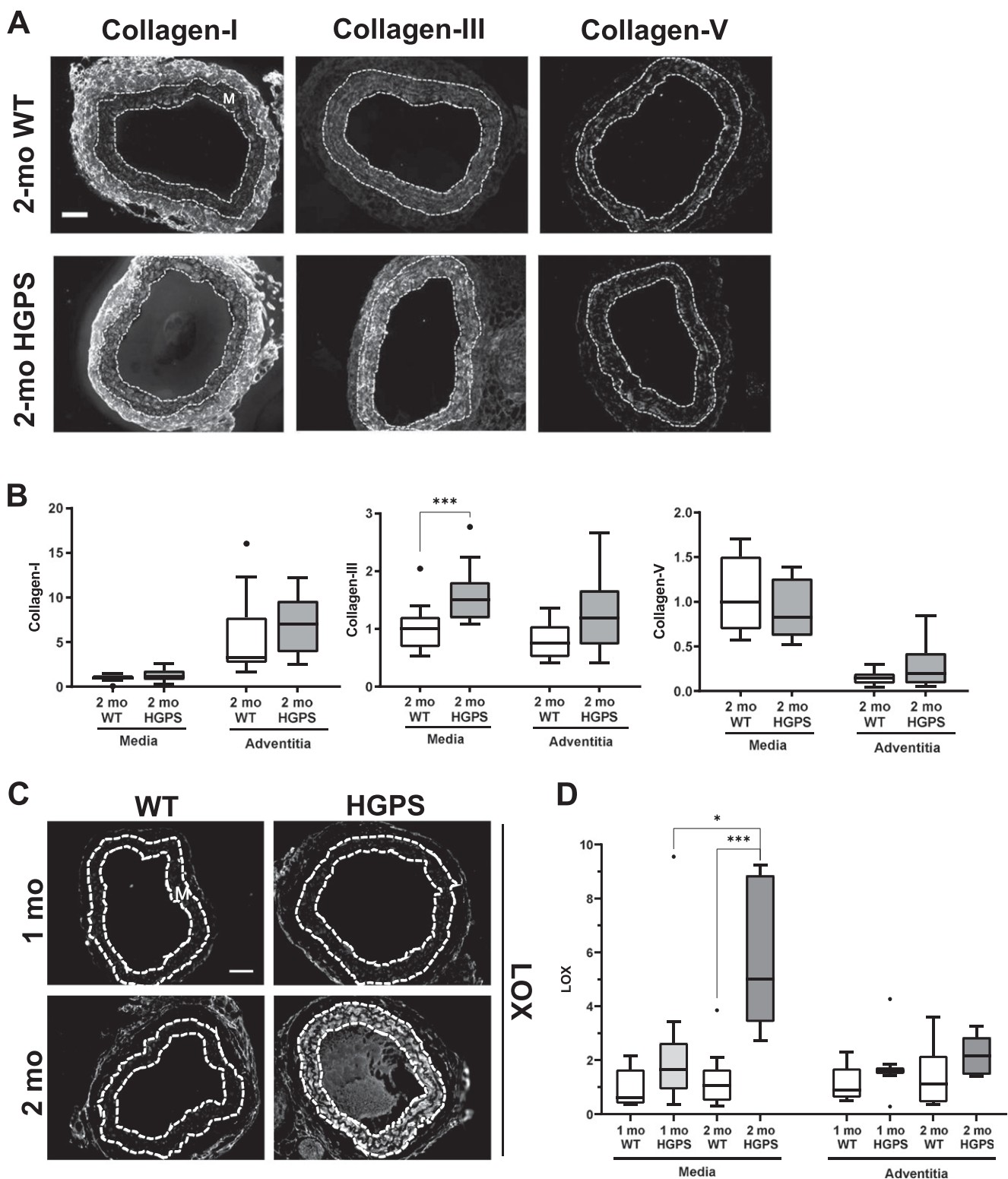

**Figure 3. Hutchinson–Gilford Progeria Syndrome (HGPS) carotid arteries display alterations in abundance of fibrillar collagens and lysyl oxidase (LOX).**
**(A)** Representative images of carotid artery cross sections from 2-mo WT and HGPS mice immunostained for collagen-I (n = 11–12), -III (n = 13–15), and -V (n = 9–10); scale bar = 50 μm. **(B)** Collagen signal intensities from the immunostained sections were quantified, and results were normalized to the mean signal intensity in the 2-mo WT media. **(C)** Representative images of LOX immunostaining of carotid artery sections from 1- to 2-mo WT and HGPS mice (n = 7–9 mice per age and genotype). Scale bar = 50 μm. **(D)** LOX signal intensities from the medial and adventitial layers were quantified, and the results were normalized to the mean signal intensity of the 1-mo WT medial layer. Statistical significance in (B) and (D) was determined by Mann–Whitney tests. The arterial media (M) is outlined with dashed lines.

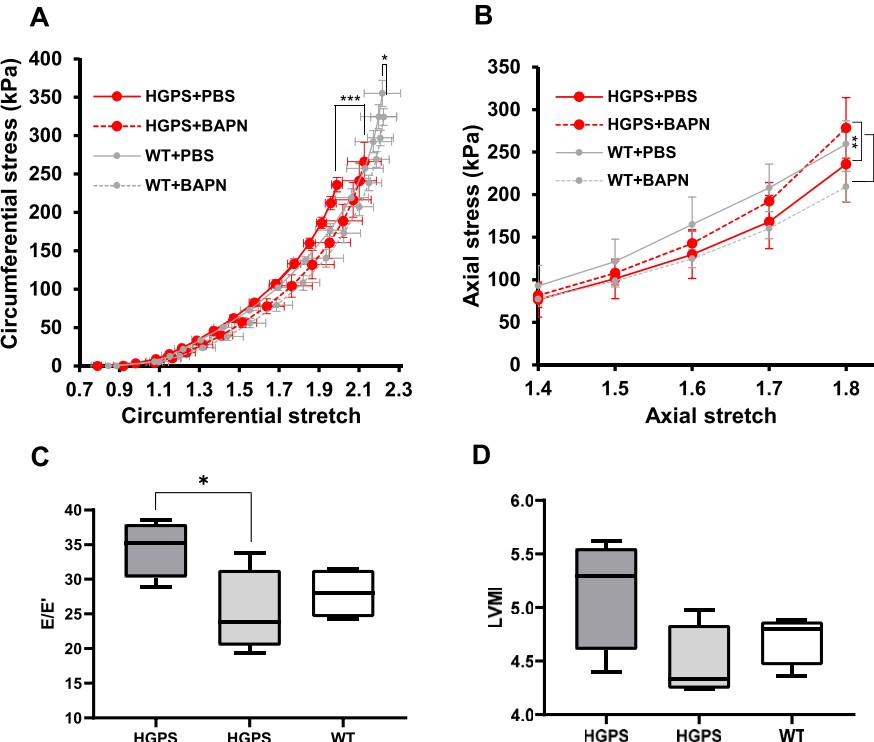

**Figure 4. Elevated lysyl oxidase expression is linked to premature arterial stiffening and diastolic dysfunction in Hutchinson–Gilford Progeria Syndrome (HGPS) mice.**
Mixed sex WT and HGPS mice aged to 1-mo were treated with β-aminopropionitrile, BAPN (WT n = 5, HGPS n = 6) or PBS (WT n = 6, HGPS n = 7) for ~30 d followed by pressure myography of isolated carotid arteries. **(A, B)** Circumferential and axial stretch–stress curves, respectively, of WT and HGPS mice treated with PBS or BAPN. Results in panels (A) display means ± SE, and results in panel (B) display means ± SD. Statistical significance for the myographic analysis was determined by two-way ANOVA. **(C, D)** Echocardiography measurements of (C) the E/E' ratio and (D) the left ventricular mass index (left ventricular mass to body weight ratio) of 64–75 d old HGPS mice (n = 4), 65–75 d old HGPS mice treated with BAPN (n = 4), and 66–75 d WT mice (n = 4). Mouse numbers in (C, D) were accrued from vehicle-injected and uninjected mice, and statistical significance was determined by Mann–Whitney tests.

to identify 852 regulators predicted to be either activated or inhibited based on the entire set of DEGs (Fig 6A and Table S2). Within this list of 852, we focused on the 139 categorized as transcription regulators (which include both transcription factors and epigenetic modifiers) and 10 categorized as microRNAs (miRs) (Fig 6A and B).

The set of 139 activated or inhibited transcription regulators identified from genome-wide analysis was compared with the IPA list of 29 transcription regulators reported to be upstream of LOX (Fig 6B; left). This analysis identified eight candidate transcription regulators of LOX (Fig 6B and Table S3) that were either activated or inhibited in HGPS. However, none of these were strong candidates for LOX mRNA regulation either because their predicted activation/ inhibition state did not match with the up-regulation of LOX mRNA in HGPS or because their reported transcriptional effects were inconsistent with the selective up-regulation of the LOX isoform we detected in HGPS (Fig 5F and G and Table S3).

We then examined the 10 upstream microRNA (miR) regulators identified by IPA (Fig 6B; right). All of these miRs were predicted to have an inhibited activation state (Table S2), but an analysis of LOX-

targeting miRs with TargetScan indicated that only one of them, miR-145, had a highly conserved target sequence in the 3' UTR of LOX (Fig 6C). The miR-145 target sequence is also absent from the LOX family members that are not strongly up-regulated in HGPS. RT-qPCR extended the IPA prediction of reduced miR-145 signature and showed that miR-145 levels are down-regulated in the medial layer of 2-mo HGPS aortas (Fig 6D) but not in the adventitial layer (Fig 6E), which (as with the protein; refer to Fig 5A and B) expressed lower levels of LOX. This down-regulation of medial miR-145 may be specific to the HGPS vasculature as it was not detected in two other SMC-containing tissues, intestine and bladder (Fig S12C and D). The levels of miR-145 were also similar in the aortas of 1-mo WT and HGPS mice (Fig S12A), consistent with the absence of strongly up-regulated LOX gene expression (Fig S12A) or arterial stiffening (Fig S6) at this early 1-mo time-point.

We were able to extend our results to isolated aortic SMCs. Primary SMCs from HGPS mice also displayed reduced miR-145 transcript levels (Fig 7A). Moreover, ectopic expression of miR-145 in primary WT and HGPS aortic SMCs reduced the level of LOX mRNA,

**Table 2. In Vivo Stretch (IVS) values for mixed sex, 2-mo WT, and Hutchinson–Gilford Progeria Syndrome (HGPS) mice treated with vehicle or BAPN.**

| Genotype | BAPN | IVS (mean) | IVS (SD) | n | *P* (relative to untreated mice) |
|----------|------|-----------|----------|---|----------------------------------|
| WT | | 1.85 | 0.05 | 6 | Reference |
| WT | ✓ | 1.84 | 0.06 | 5 | NS |
| HGPS | | 1.73 | 0.07 | 7 | Reference |
| HGPS | ✓ | 1.74 | 0.05 | 6 | NS |

Significance was determined by Mann–Whitney test. NS; not significant.

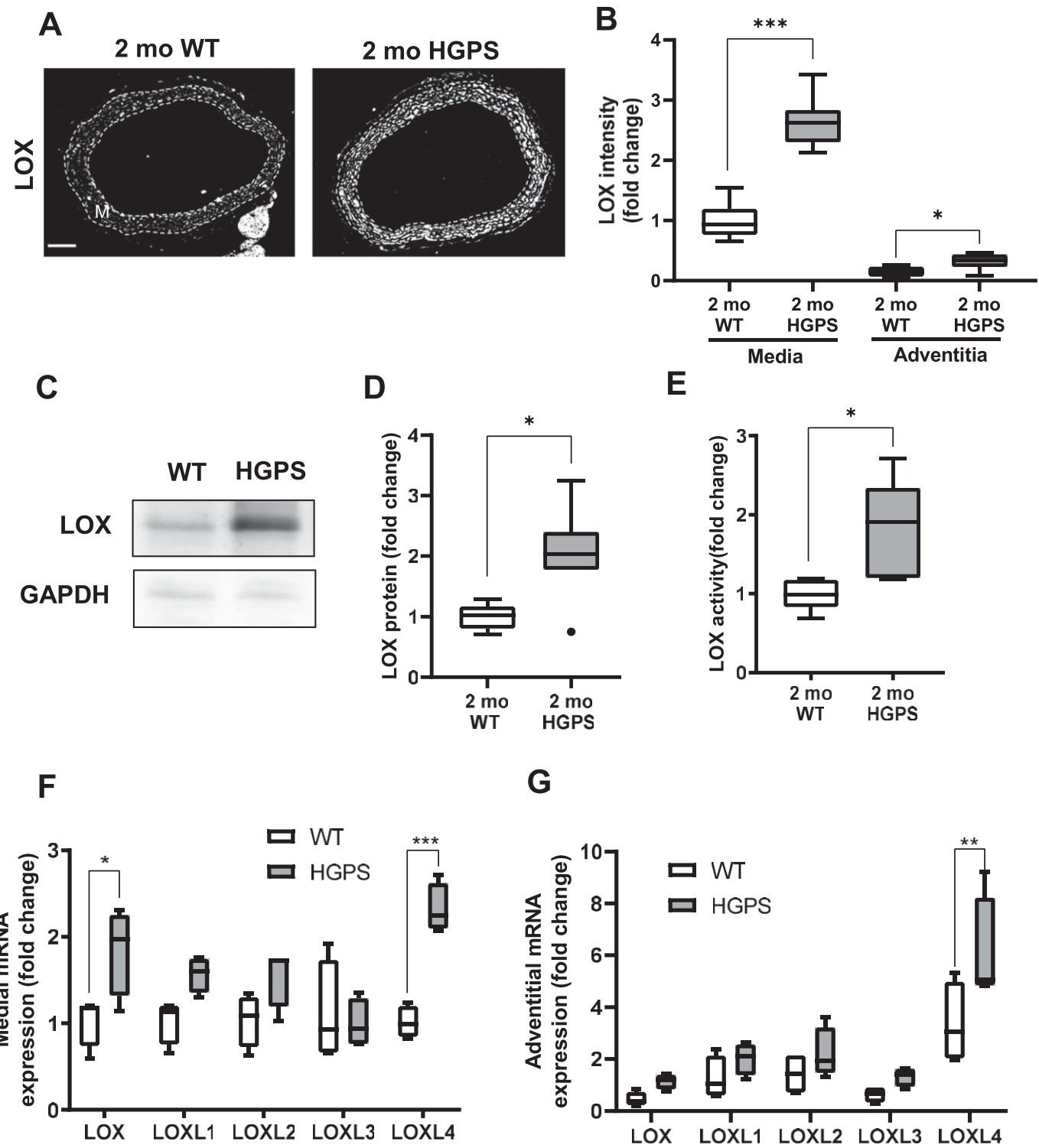

**Figure 5. Hutchinson–Gilford Progeria Syndrome (HGPS) aortas display increased lysyl oxidase (LOX) abundance and activity.**
**(A)** Representative images of 2-mo WT (n = 8) and HGPS (n = 7) aortic sections immunostained for LOX; scale bar = 100 μm. The arterial media (M) is outlined with dashed lines. **(B)** LOX signal intensities from the immunostained sections were quantified, and results were normalized to the mean signal intensity of the 2-mo WT medial layer. **(C, D)** Representative immunoblot of aortic lysate from 2-mo WT and HGPS mice, and (D) quantification of immunoblots (n = 6–7 per genotype). **(E)** LOX activity in 2-mo WT and HGPS aortic tissue (n = 5 per genotype). Statistical significance in A-E was determined by Mann–Whitney tests between genotypes. **(F, G)** mRNA expression levels of LOX family isoforms in adventitia-free 2-mo WT and HGPS aortas and (G) 2-mo WT and HGPS aortic adventitial tissue were quantified by RT-qPCR (n = 4 independent experiments, with two aortas pooled per experiment). Transcript levels were normalized to 2-mo WT medial layer values. Statistical significance was determined by two-way ANOVA followed by Holm–Sidak post-tests.

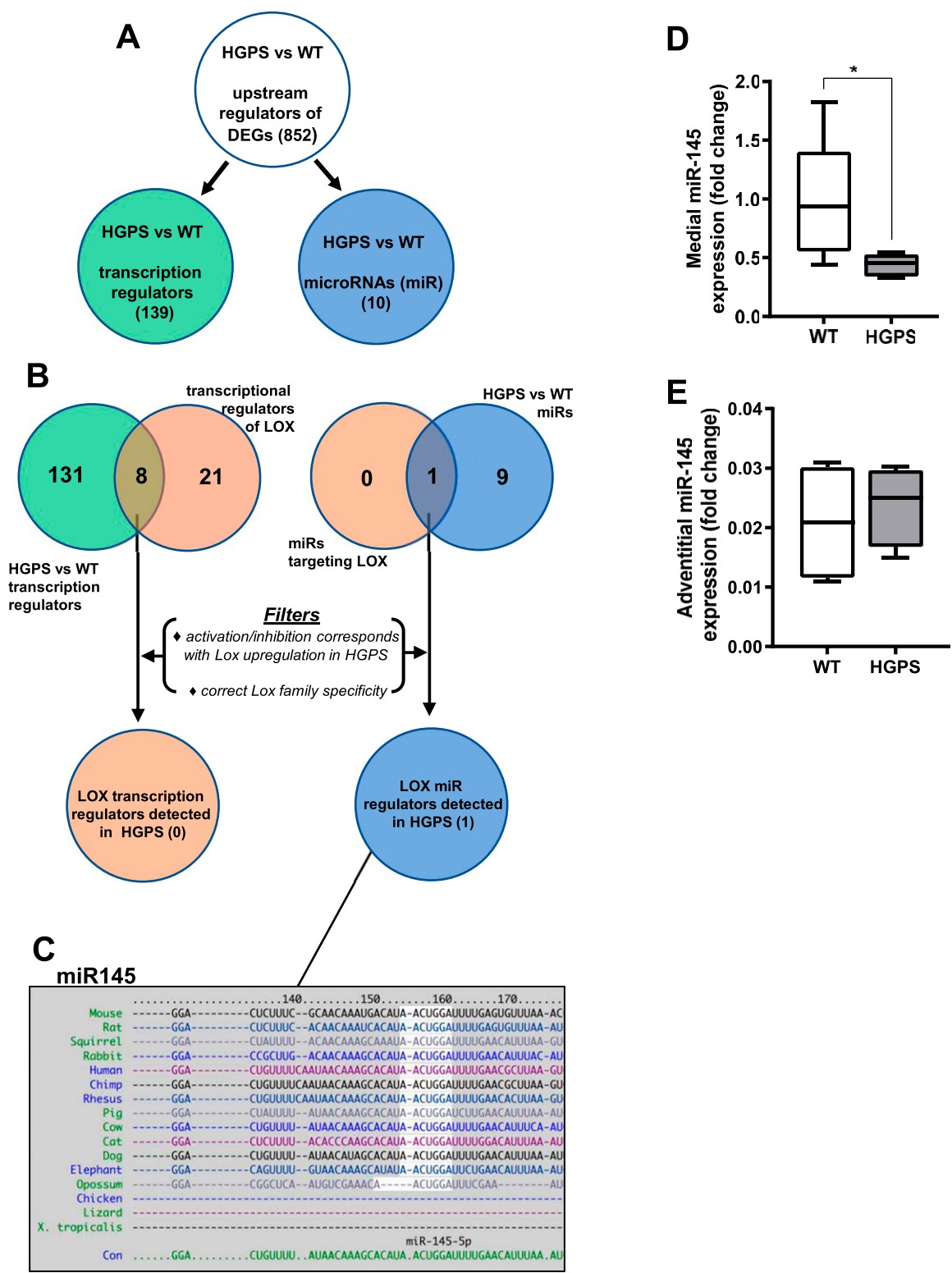

**Figure 6. Hutchinson–Gilford Progeria Syndrome (HGPS) aortic tissues show reduced miR-145 transcript levels.**
RNA-sequencing analysis was performed (see the Materials and Methods section) on cleaned aortas from 2-mo WT and HGPS mice (n = 6 per genotype) to identify potential upstream regulators of LOX. **(A)** The number of activation or inhibition signatures in the HGPS versus WT aortas was inferred from an Ingenuity Pathway Analysis of differentially expressed genes, and the list of total activation/inhibition signatures (top) was subdivided into transcription regulators and microRNAs. **(B)** Venn diagram identifying the number of activation/inhibition signatures in HGPS categorized as either transcription regulators (left, green) or microRNAs (right, blue) were compared with known transcriptional or microRNA regulators of LOX identified from Ingenuity Pathway Analysis (beige). The likelihood of the eight transcription regulators and one miR accounting for LOX up-regulation in HGPS was then considered individually using the criteria shown ("Filters"). **(C)** TargetScan depiction of the conserved 3′ UTR miR-145-5p target sequence in LOX. **(D, E)** miR-145 transcript levels from the medial layer (n = 6) and (E) adventitial layer (n = 4) of 2-mo WT and HGPS mice. miR-145 transcript levels were normalized to WT media values. Statistical significance was determined by Mann–Whitney tests.

and the elevated expression of LOX mRNA seen in HGPS SMCs became similar to that of the WT controls (Fig 7B). Although these results do not exclude a possible role for aberrant transcriptional control of LOX in HGPS, they strongly support miR-145 down-regulation as an important and causal regulator of arterial LOX abundance in HGPS.

As HGPS is associated with reduced expression of WT Lamin A as well as expression of progerin (see Introduction), we depleted Lamin A from WT SMCs with siRNAs and asked if reduced Lamin A abundance would be sufficient to generate the miR-145/LOX relationship seen in HGPS. Distinct Lamin A siRNAs effectively reduced Lamin A protein levels (Fig 7C) but did not significantly alter the expression levels of LOX protein (Fig 7C), LOX mRNA (Fig 7D), or miR-145 (Fig 7D). We conclude that a progerin-mediated gain-of-function in HGPS mice is responsible for the down-regulation of miR-145 and the consequent increase of LOX gene expression in HGPS.

Finally, we used TargetScan to identify several other putative miR-145 targets within the DEGs of 2-mo WT and HGPS aortas (Fig S14A). We selected three of those genes based on fold changes that were greater or equal to that of LOX and then examined changes in their mRNA levels in isolated SMCs with and without ectopic miR-145 expression. None showed the strong regulation by miR-145 seen with LOX (Fig S14B). Thus, arterial LOX mRNA appears to be a particularly strong miR-145 target in HGPS.

### Distinct regulation of the ECM, lysyl oxidase, and miR-145 in HGPS and normal aging

As many parallels have been drawn between HGPS and normal aging, we evaluated levels of the fibrillar collagens, LOX, and miR-145 in aged (24-mo) WT mice (Fig 8). The carotid arteries of the aged WT mice showed only small changes in medial collagen III and collagen-V (Fig 8A and B). However, aged WT mice showed a statistically significant increase in adventitial collagen-I (Fig 8A and B); this was not seen in HGPS (refer to Fig 3). LOX protein increased with time in both the carotid medial and adventitial layers during normal aging (Fig 8C and D), whereas it was mostly limited to the carotid media in HGPS (refer to Fig 3C and D). Unexpectedly, our RNASeq analysis indicated that LOX mRNA abundance was minimally altered in the aortas of 2- versus 24-mo WT mice (Fig S13C), and RT-qPCR showed that this change was not significant (Fig 8E). Similarly, arterial miR-145 levels were not decreased in aged WT mice (Fig 8F). These data indicate that arterial LOX up-regulation in normal aging is mechanistically distinct from that seen in HGPS. Thus, the down-regulation of miR-145 and consequent steady-state increase in LOX mRNA and protein we describe in HGPS are disease-specific effects.

## Discussion

We show here that the carotid arteries of HGPS mice stiffen prematurely and that this stiffening is preferentially circumferential, particularly in males. Although the expression of several arterial collagens is elevated in old HGPS mice (del Campo et al, 2019), our results show that the initiation of arterial stiffening in HGPS is much more closely associated with increased medial LOX expression. Importantly, pharmacologic intervention with the pan-LOX inhibitor, BAPN, improved arterial mechanics of HPGS mice, thereby establishing LOX up-regulation as an underlying and causal mechanism in the early arterial stiffening of HGPS.

Although we did not see pronounced increases in arterial fibrillar collagens early in HGPS, TEM revealed a slight increase in

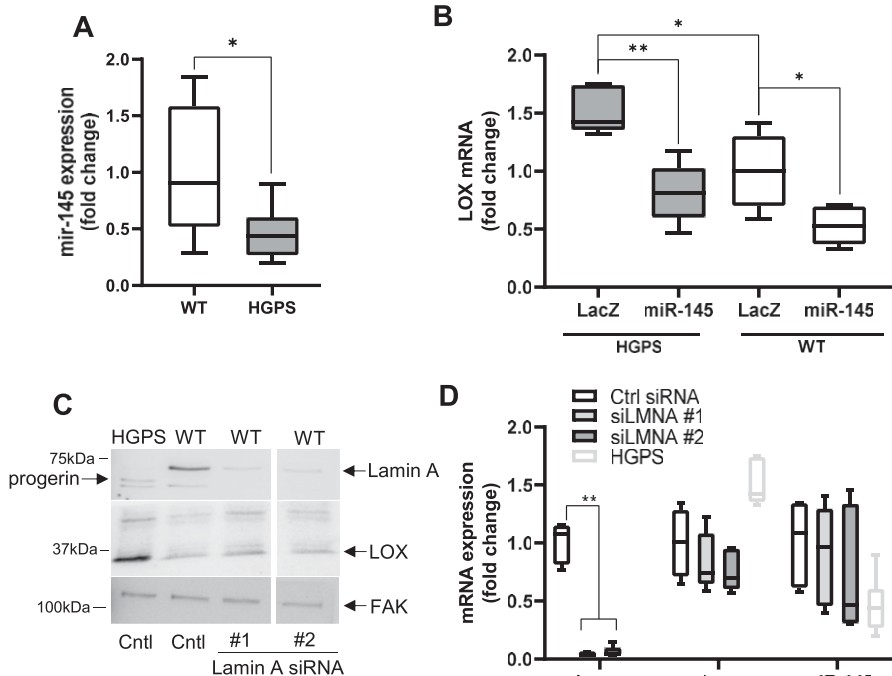

**Figure 7. Ectopic expression of miR-145 reduces lysyl oxidase (LOX) mRNA in primary Hutchinson–Gilford Progeria Syndrome (HGPS) vascular smooth muscle cells (SMCs).**
**(A)** miR-145 transcript levels in isolated WT and HGPS SMCs as determined by RT-qPCR (n = 8 per genotype). Statistical significance was determined by a Mann–Whitney test. **(B)** LOX mRNA levels in WT and HGPS SMCs infected with adenoviruses encoding LacZ (control) or miR-145 determined by RT-qPCR and normalized to the WT LacZ control (n = 5). Statistical significance was determined by Mann–Whitney tests. **(C)** WT SMCs were transfected with a control siRNA or two different siRNAs targeting LMNA; lysates were analyzed by Western blotting and probed for Lamin A, LOX, and FAK (loading control) (n = 3). HGPS SMCs were included as reference. White spaces indicate removal of extraneous data. **(D)** LMNA mRNA was knocked-down in WT SMCs using two distinct siRNAs, and transcript levels of Lmna, Lox, and miR-145 were analyzed by RT-qPCR and normalized to the Ctrl siRNA treatment (n = 5). Statistical significance was determined by two-way ANOVA with Holm-Sidak post-tests. The light gray bars in the Tukey plots show HGPS references for miR-145 and LOX and are reproduced from Fig 7A and B, respectively.

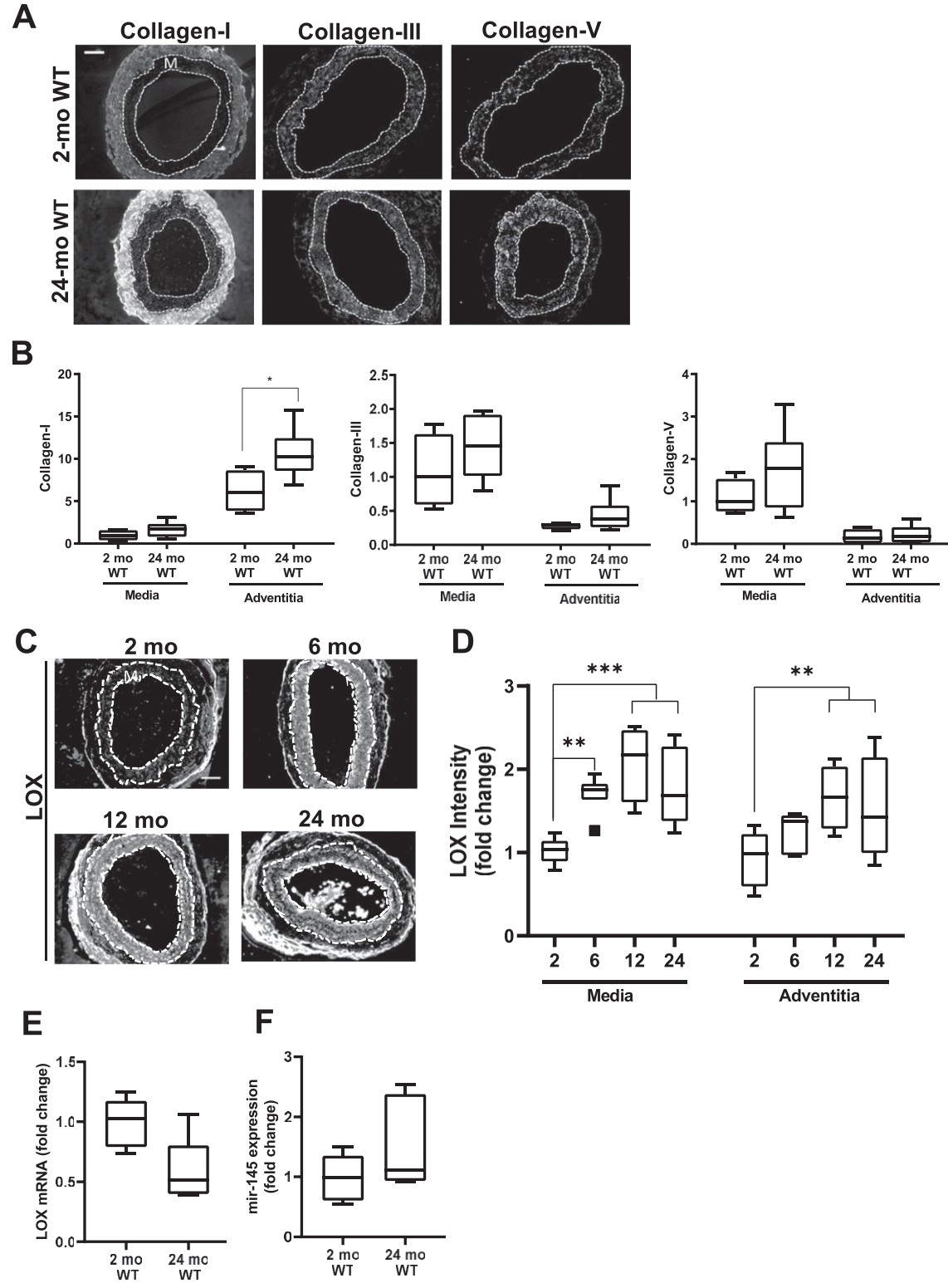

**Figure 8. Differential mechanisms drive overexpression of lysyl oxidase (LOX) in normal aging.**

C57BL/6 (WT) mice were aged from 2 to 24 mo. **(A)** Representative images of collagen-I, collagen-III, collagen-V immunostaining in carotid artery sections of 2- and 24-mo WT mice (n = 4–7 mice per age-group). **(B)** Collagen signal intensities in the medial and adventitial layers of the immunostained carotid cross sections were quantified, and results were normalized to the mean of the 2-mo WT medial layer for each collagen. Statistical significance was determined by Mann–Whitney tests between ages. **(C)** Representative images of LOX immunostaining of carotid artery cross sections from 2-mo (n = 8), 6-mo (n = 7), 12-mo (n = 4), and 24-mo (n = 9) WT mice. **(D)** LOX signal intensities were quantified from the immunostained sections, and results were normalized to the mean signal intensity of the 2-mo WT medial layer. Statistical significance was determined by one-way ANOVA relative to the 2-mo mice followed by Holm–Sidak post-tests. **(E, F)** LOX mRNA (n = 5 per age) and (F) miR-145 expression levels (n = 6 per age) in 2- and 24-mo WT aorta determined by RT-qPCR and normalized to 2-mo WT. Statistical significance was determined using Mann–Whitney tests. In carotid cross section images, the arterial media (M) is outlined with dashed lines; scale bar = 50 $\mu$m.

collagen area within the medial elastin folds of HGPS arteries. As adventitial collagen abundance was similar in WT and HGPS mice, the medial collagen in these folds may be the target of the elevated medial LOX in HGPS. Indeed, previous studies have reported that the medial layer contributes more toward circumferential stiffening, whereas the adventitial layer contributes more to axial mechanics (Kohn et al, 2015). Because the expression of collagen genes can increase with ECM stiffness (Kothapalli et al, 2012), the LOX-mediated increase in arterial stiffness described in our studies here with very young HGPS mice may also contribute to the eventual induction of other SMC collagens as seen when HGPS mice are aged (del Campo et al, 2019).

In addition to reducing arterial stiffness, BAPN administration also improved diastolic function in HGPS mice. Diastolic dysfunction is a prominent cardiac abnormality in both HGPS children (Prakash et al, 2018) and HGPS animal models (Osmanagic-Myers et al, 2018; Dorado et al, 2019; Murtada et al, 2020). Although the basis of diastolic dysfunction is multi-faceted, it has been correlated with increased arterial stiffness (Mottram et al, 2005; Kim et al, 2017). Thus, the reduction in arterial stiffness by BAPN may be causal for the observed improvement in diastolic function. However, it is not yet clear if our systemic administration of BAPN is affecting diastolic dysfunction indirectly through its effects on arterial stiffness, or directly through effects on LOX activity in the heart. A comparative analysis of BAPN effects on arterial versus cardiac (cardiomyocyte and cardiac fibroblast) LOX abundance and enzymatic activity, as well as the consequences of BAPN treatment for the arterial and cardiac ECMs, will be required to address this issue.

Genome-wide analysis predicted that the elevated expression of LOX in HGPS is linked causally to an abnormal down-regulation of miR-145 in vascular SMCs. Others have shown that miR-145 is one of the most highly expressed microRNAs in rodent carotid arteries, where it is largely restricted to the SMCs (Cheng et al, 2009; Zhang, 2009). miR-145 is thought to play a critical role in vascular smooth muscle function and a protective role preventing fibrotic ECM synthesis (Zhao et al, 2015). In addition, Faccini and colleagues (Faccini et al, 2017) identified reduced circulating levels of miR-145 as a diagnostic biomarker for coronary artery disease. Our results extend those findings to HGPS, which shows reduced aortic miR-145 abundance relative to age-matched WT arteries and isolated SMCs.

In addition to the bioinformatic prediction, a connection between miR-145 and LOX mRNA abundance has been described experimentally. We have previously reported that overexpression of miR-145 reduced LOX mRNA levels in WT SMCs (Kothapalli et al, 2012). In addition, the same study showed that a miR-145 antagomir counteracted the suppression of LOX mRNA seen in WT SMCs having elevated miR-145. The work described here extends these studies by demonstrating that enforced expression of miR-145 restores near-normal LOX mRNA levels to HGPS SMCs. Collectively, these results strongly suggest a causal connection between the decreased expression of miR-145 and the increased expression of LOX in HGPS. A reporter assay showing that miR-145 can regulate mRNA abundance through the LOX 3′ UTR would further support this connection.

Reduced expression of WT Lamin A failed to regulate miR-145 or LOX mRNA, indicating that misregulation of miR-145 and LOX in HGPS reflects a gain-of-function from progerin expression. However, how progerin may be eliciting these effects is cause for further

study. As it has recently been appreciated that Lamin A and the nuclear lamina play an important role in mediating transfer of information from the cytoplasm to the nucleus, the altered nuclear structure due the presence of progerin in HGPS may hinder or promote transfer of this information across the nuclear boundary, leading to deregulated gene expression (Kelley et al, 2011; Wilson, 2018). In addition, progerin may have a more direct effect on miR-145 expression, as recent studies have shown that HGPS nuclei have a dramatically altered epigenetic signature which can lead to improper gene silencing or activation (Arancio et al, 2014; Köhler et al, 2020). Altered LOX and miR-145 expression in HGPS could also be a secondary effect of long-term progerin expression, perhaps with progerin-expressing SMCs secreting altered chemokines to promote vascular remodeling in HGPS.

As progerin has been detected in aged tissues, cells, and atherosclerotic lesions (Scaffidi & Misteli, 2006; McClintock et al, 2007; Olive et al, 2010), there has been significant interest in the idea that progerin-like splicing may contribute to phenotypes of natural aging in addition to driving early aging in HGPS. Indeed, both 2-mo HGPS and 24-mo WT carotid arteries display increased expression of medial LOX and increased circumferential stiffness. However, the axial mechanics of 2-mo male HGPS carotid arteries are mostly intact, whereas 24-mo male WT carotids display clear axial stiffening. This distinction implies that in addition to the common effect of increased LOX abundance, there must also be inherent differences between the early arterial stiffening in HGPS and the late arterial stiffening of normal aging. Interestingly, this distinction in the directionality of arterial stiffening is lost as HGPS progresses; HGPS mice much nearer to the end of their life display axial stiffening (Murtada et al, 2020).

Others have recently reported increased LOX and LOXL2 in old (20–22 mo) versus young (3–4 mo) aortas of WT mice (Steppan et al, 2019). This work complements our findings of increased LOX expression in HGPS carotid arteries, and the joint findings support the idea that increased LOX activity with age is likely to be a general feature of large artery stiffening. However, in our work, LOX, rather than LOXL2, is the major up-regulated arterial LOX family member. We also identified up-regulated levels of LOXL4 mRNA in HGPS. As BAPN is an inhibitor of all LOX isoforms, we cannot exclude a potential role for LOXL4 in arterial stiffening. However, LOXL4 expression levels were much lower than the LOX isoform.

Our immunofluorescence analysis shows that LOX is induced in both the arterial medial and adventitial layers with natural aging whereas its induction is largely restricted to the media in HGPS. Finally, unlike HGPS, the levels of miR-145 are not strongly reduced in aged WT mice. The preferential expression of miR-145 by arterial SMCs (Cheng et al, 2009; Zhang, 2009) could explain why we see a selective effect on medial rather than adventitial LOX in HGPS. Thus, whereas an increase in overall LOX activity is a common feature of arterial stiffening in both natural aging and HGPS, the uncoupling of LOX expression from miR-145, the distinct spatial expression patterns of arterial LOX, and the differences in directional arterial stiffening distinguish natural aging and HGPS. Although we show that LOX levels are not increased in other SMC-containing tissues in HGPS mice (i.e., bladder and intestine), it would be of interest to evaluate LOX and miR-145 expression levels in non–SMC-containing tissues such as the skin, which also displays abnormalities in HGPS (Gonzalo et al, 2016).

Administration of sodium nitrate to HGPS mice (del Campo et al, 2019) and administration of Lonafarnib, a farnesyltransferase inhibitor, to HGPS children reduce arterial stiffness (Gordon et al, 2012, 2018). Furthermore, the reduction in arterial stiffness by lonafarnib was associated with an increased lifespan of HGPS children (Gordon et al, 2018). These studies did not distinguish between effects on axial versus circumferential mechanics, nor are the mechanisms by which nitrate and lonafarnib reduce arterial stiffness fully clear. Nevertheless, these studies and this report collectively indicate that decreasing arterial stiffness could lessen the burden of cardiovascular disease in HGPS. Our results further indicate that targeting circumferential arterial mechanics, potentially though LOX inhibition, may be an important consideration in the development of mechanically inspired therapeutics for HGPS.

## Materials and Methods

### Mice and artery isolation

WT C57BL/6 mice were purchased from Jackson Labs and aged to 24-mo. LMNA$^{G609G/+}$ mice on the C57BL/6 background were generously provided Dr. Carlos Lopez-Otin (Universidad de Oviedo). Mice were genotyped using the following primers: forward: AAGGGGCTGGGAG-GACAGAG; and reverse: AGCATGCCATAGGGTGGAAGGA. Mice were fed a chow diet ad libitum. At the appropriate age, the mice were euthanized by $CO_2$ asphyxiation, the left carotid artery was immediately removed, stripped of most fat, and used for pressure myography as outlined below. The remaining arteries were perfused in situ with PBS. The right carotid artery was then removed, cleaned in PBS and fixed in either Prefer for paraffin-embedding or TEM fixative (see below). The descending aorta was isolated from the end of the aortic arch to the diaphragm, cleaned as above, and used either for the preparation of RNA (for RT-qPCR and genome-wide analysis) or protein for immunoblotting (see below and Supplemental Data 1). Animal protocols were approved by the University of Pennsylvania Institutional Animal Care and Use Committee. All experiments were performed on male mice unless otherwise specified in the legends.

### Biaxial extension–inflation tests using a pressure myograph

Arterial mechanics were determined on a DMT 114P pressure myography with force transducer largely as described (Brankovic et al, 2019). Freshly isolated carotid arteries from WT and HGPS mice (with a mean age of 31 ± 1 d for 1-mo mice, 64 ± 6 d for 2-mo mice, and 726 ± 9 d for 24-mo mice) were secured to 380 $\mu$m (outer diameter) cannulas using silk sutures; blood was cleared, and any remaining fat was removed. Once mounted, the arteries were visualized by light microscopy, and the unloaded/unpressurized arterial wall thickness and inner radius was measured at the axial length where the artery transitioned from being bent to straight. Arteries were brought to a stretch of 1.7 and pressurized to 100 mm Hg for 15 min with HBSS. The arteries were then preconditioned by cyclic pressurization three times from 0 to 140 mm Hg in 1-min increments. Unloaded (unstretched and unpressurized) vessel wall

thickness and outer diameter were measured in multiple sections after preconditioning and averaged for post-test data analysis.

IVS was determined using force-length tests as described (Ferruzzi et al, 2013; Brankovic et al, 2019). Briefly, the carotid arteries were axially stretched in 10% increments at three constant pressures (90, 120, and 140 mm Hg). Equilibrium force was recorded for each stretch and pressure, and the intersection of the three force–stretch curves was defined as the IVS. Loaded inner radius and wall thickness were determined from pressure-outer diameter tests with samples at their IVS and pressurized in 10-mm Hg (30-s) steps from 0 to 140 mm Hg before returning the artery to 0 mm Hg (Brankovic et al, 2019). This test was performed three times, and the mean of three stress–stretch curves was taken. We confirmed the validity of our IVS determinations by measuring axial force through the circumferential tests, and we excluded samples where axial force varied from the mean by >25% with pressure. Stress–stretch relationships were also analyzed by deriving the tangent modulus. See Supplemental Data 1 for further details of data analysis.

### TEM analysis of collagen structure

Carotid arteries for TEM were fixed overnight at 4°C in 0.1M sodium cacodylate buffer, pH 7.4 containing 2.5% glutaraldehyde and 2.0% paraformaldehyde. After subsequent buffer washes, the samples were post-fixed in 2.0% osmium tetroxide for 1 h at room temperature and rinsed in water before en bloc staining with 2% uranyl acetate. Briefly, after dehydration through a graded ethanol series, the tissue was infiltrated and embedded in EMbed-812 (Electron Microscopy Sciences). Thin cross sections were stained with uranyl acetate and lead citrate and examined with a JEOL 1010 electron microscope fitted with a Hamamatsu digital camera and AMT Advantage image capture software. Images of artery cross sections were taken at increasing magnification; collagen abundance was evaluated as described in Supplemental Data 1.

### In vivo treatment with BAPN

Mixed sex WT and HGPS mice (a mean age of 35 ± 2 d old) were injected peritoneally with BAPN (A3134; 333 mg/kg; Sigma-Aldrich) or an equal volume of vehicle (PBS). BAPN was dissolved in PBS and injected in a volume of 0.2 ml per day until the mice reached 2-mo of age (a mean of 24 ± 2 d of injections). No dramatic alterations in mouse behavior, no changes in appearance, and no weight loss were observed during the injection period. At ~2-mo of age (a mean of 60 ± 2 d old), the mice were euthanized, carotid arteries were isolated, and unloaded wall thickness and inner radii were determined. The samples were then analyzed by pressure myography as described above. For the echocardiography and invasive hemodynamics experiments, male mice were analyzed between 64 and 75 d of age immediately after a 30-d treatment with BAPN (333 mg/kg as described; [Kothapalli et al, 2012]). Echo and hemodynamics performed by the UPenn Cardiovascular Phenotyping Core. See Supplemental Data 1 for additional echocardiography methods.

### Genome-wide sequencing and analysis

Descending aortas containing the intimal, medial, and adventitial layers were isolated from six 2-mo male WT, six 24-mo WT mice, and

six HGPS mice, and RNA was extracted using the RNeasy Plus Micro kit (74034; QIAGEN). The high-throughput library was prepared using the truSeq stranded total RNA (ribo-Zero) kit (20037135; Illumina). Paired end sequencing was performed on a HiSeq4000 Sequencing System (Illumina) and generated 14–30 million reads/sample.

Raw sequence files were mapped to the genome with using salmon (https://combine-lab.github.io/salmon/) against the mouse transcripts described in genecode (version M23, built on the mouse genome GRCm38.p6, https://www.gencodegenes.org). Transcript counts were summarized to the gene level using tximport (https://bioconductor.org/packages/release/bioc/html/tximport.html), and normalized and tested for differential expression using DESeq2 (https://bioconductor.org/packages/release/bioc/html/DESeq2.html). Sequence files can be found under GEO ascension number GSE165409. The normalized values for the 300 genes with the highest variance across all samples were imported into Partek Genomics Suite (v7, Partek, Inc.) for principal component analysis.

DEGs having a 1.5× fold change and adjusted $P$-values of <0.001 were subjected to gene ontology analysis (GO cellular component) using DAVID (https://david.ncifcrf.gov/home.jsp) and a core analysis with IPA. Default settings were used for all other core analysis parameters. The core analysis included the prioritization of upstream regulators based on enrichment of a regulator's target gene set in the set of DEGs. For each regulator, IPA tested the directionality of each of the genes in the overlap to infer an activation or inactivation of the regulator (Z-score). We defined putative upstream regulators of DEGs as those having Z-scores >1.75 (activated) or <−1.75 (inhibited). This set of upstream regulators was then filtered to identify those classified by IPA as either transcription regulators (including transcription factors and epigenetic regulators) or microRNAs (miRs).

The list of transcription regulators categorized by IPA as either being in an activated or inhibited state (positive or negative Z-scores, respectively) was compared with the transcriptional regulators of LOX present in the IPA database. The overlapping gene set was individually examined to determine if its proposed activation or inhibition state correctly corresponded to the experimental up-regulation of LOX we observed in HGPS. We also checked for the specificity of upstream regulator effect on the LOX gene. These comparisons allowed us to assess the appropriateness of the IPA-proposed transcription regulator relationship. The list of 10 miRs identified by IPA is categorized as being in an inhibited activation state in HGPS (negative Z-scores). The entire miR list was therefore compared with the list of miRs targeting mouse LOX as determined by TargetScan (www.targetscan.org).

### Statistical analysis

Statistical analysis was performed using Prism software (GraphPad). For the pressure myography experiments, differences in inner radius, wall thickness, axial stress and circumferential stretch at pressure-matched points were compared by age, genotype, or response to BAPN across the entire curves and analyzed by two-way ANOVA relative to the relevant 2-mo WT control. For other mouse experiments and for all studies with isolated cells, ANOVAs were used for multiple comparisons, and two-tailed Mann–Whitney tests were used to compare two datasets. Statistical significance for all graphs is demarcated by *($P$ < 0.05), **($P$ < 0.01), ***($P$ < 0.001). Box plots show Tukey whiskers.

## Data Availability

The expression profiling by high-throughput sequencing from this publication has been deposited to the GEO database (https://www.ncbi.nlm.nih.gov/geo) and assigned the identifier GSE165409.

## Supplementary Information

## Acknowledgements

We thank Dr. Carlos Lopez-Otín for the generous gift of the LMNA[G609G] mouse line and the Electron Microscopy Research Laboratory at the University of Pennsylvania for preparing arterial tissue for TEM, the Next Generation Sequencing Core at University of Pennsylvania for performing the RNASeq, and the Mouse Cardiovascular Phenotyping Core at University of Pennsylvania for mouse echocardiography. This work was supported by National Institutes of Health grants AG047373 and AG062140, the Progeria Research Foundation, and the Center for MechanoBiology, a National Science Foundation Science and Technology Center under grant agreement CMMI: 15-48571. R von Kleeck was supported by NIH grants T32-GM008076 and F31-HL142160.

### Author Contributions

R von Kleeck: conceptualization, data curation, formal analysis, methodology, and writing—original draft, review, and editing.
E Roberts: data curation and formal analysis.
P Castagnino: conceptualization, data curation, investigation, and methodology.
K Bruun: data curation, formal analysis, and methodology.
SA Brankovic: data curation and formal analysis.
EA Hawthorne: data curation and methodology.
T Xu: data curation.
JW Tobias: formal analysis.
RK Assoian: conceptualization, data curation, formal analysis, supervision, funding acquisition, methodology, and writing—original draft, review, and editing.

### Conflict of Interest Statement

The authors declare that they have no conflict of interest.

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
