## [Reviewer comments · Life Science Alliance]

Life Science Alliance

Arterial stiffness in Hutchinson-Gilford Progeria Syndrome corrected by inhibition of Lysyl Oxidase

Ryan von Kleeck, Emilia Roberts, Paola Castagnino, Kyle Bruun, Sonja A Brankovic, Elizabeth A Hawthorne, Tina Xu, John W Tobias and Richard Assoian

DOI: <https://doi.org/10.26508/lsa.202000997>

Corresponding author(s): Richard Assoian (University of Pennsylvania) and Ryan Von Kleeck (University of Pennsylvania)

Review Timeline:	Submission Date:	2020-12-17
	Editorial Decision:	2021-02-02
	Revision Received:	2021-02-12
	Accepted:	2021-02-23

Scientific Editor: Shachi Bhatt

Transaction Report:

Please note that the manuscript was previously reviewed at another journal and the reports were taken into account in the decision-making process at Life Science Alliance. Since the original reviews are not subject to Life Science Alliance's transparent review process policy, the reports and author response cannot be published.

February 2, 2021

RE: Life Science Alliance Manuscript #LSA-2020-00997-T

Dr. Richard Assoian
University of Pennsylvania
Systems Pharmacology and Translational Therapeutics
Philadelphia, Pennsylvania 19104

Dear Dr. Assoian,

Thank you for submitting your revised manuscript entitled "Arterial stiffness and cardiac dysfunction in Hutchinson-Gilford Progeria Syndrome corrected by inhibition of Lysyl Oxidase".

For a brief overview, this manuscript was previously reviewed and accordingly revised for non-alliance journal. The authors then transferred the previous review comments, the editor's decision letters, their point-by-point rebuttal and a revised manuscript to Life Science Alliance (LSA). After a thorough review for the manuscript and the peer-review material, the manuscript was deemed to be appropriate for LSA, provided an arbitrator (referee) signs off on the revised version.

The referee's comments are copied below.

As you will note, the referee is overall enthusiastic about the manuscript. They have raised some points (cardiac dysfunction markers and miR145 loss of function analysis) that we encourage you to address only if the data is readily available. If the data is not readily available or attainable, we suggest you tone down the respective conclusions and address these issues in Discussion. The referee's request asking for better description of the RNA analysis (pt 3) in a publicly available database should be addressed.

Along with the points listed below, please also attend to the following,

- please consult our manuscript preparation guidelines <https://www.life-science-alliance.org/manuscript-prep> and make sure your manuscript sections are in the correct order
- please make sure the author order in your manuscript and our system match, and that there is no name discrepancy, i.e. names in the ms file must match with the names in the system
- please add your supplementary figure legends to the main manuscript text after the reference's section as well (please separate the Figure legends and Supplemental Figure legends into separate sections)
- please add ORCID ID for secondary corresponding author-they should have received instructions on how to do so
- please add a Category, Author Contribution, Running title, and Summary blurb for your manuscript in our system
- please add a conflict of interest statement to your main manuscript text
- please upload your main manuscript text as an editable doc file
- please upload your main and supplementary figures as single files
- please upload your tables as editable .doc or .xls file
- please add callouts for Figures S1 E,F; S3 A,B,C; S4 A,B; S7 A,B,C,D; S10 A, B,C,D,E,F to your main manuscript text

- please use the [10 author names, et al.] format in your references (i.e. limit the author names to the first 10)
- please deposit the RNA seq data into a publicly available database and provide the accession number in a 'Data Availability' section in the manuscript (<https://www.life-science-alliance.org/manuscript-prep#datadepot>)
- please provide higher resolution images for the blots in Figures 5C, 7C and S7C

A. FINAL FILES:

B. MANUSCRIPT ORGANIZATION AND FORMATTING:

****It is Life Science Alliance policy that if requested, original data images must be made available to the editors. Failure to provide original images upon request will result in unavoidable delays in publication. Please ensure that you have access to all original data images prior to final**

submission.**

The license to publish form must be signed before your manuscript can be sent to production. A link to the electronic license to publish form will be sent to the corresponding author only. Please take a moment to check your funder requirements.

Sincerely,

Shachi Bhatt, Ph.D.
Executive Editor
Life Science Alliance
<https://www.lsajournal.org/>
Tweet @SciBhatt @LSAJournal

Reviewer #1 (Comments to the Authors (Required)):

In this report, aortic stiffness was observed and characterized in HGPS and ageing mice at functional, morphological and molecular levels. Lox activation was identified as an early event and its contribution to aortic stiffening was tested using a specific Lox inhibitor. The mechanistic analysis for Lox activation was performed by transcriptome profiling which led to the finding of miR-145. While the role of miR-145 in Lox regulation was tested in SMCs, its involvement appears to be specifically limited to HGPS model rather than natural ageing.

The observation of early onset of aortic stiffening and the characterization of the underlying mechanisms in HGPS mice contributed new insights to the pathophysiology and mechanism of cardiovascular ageing due to LaminA mutation. The finding of Lox induction and miR-145 mediated regulation of Lox expression have potential value in mechanistic understanding, diagnosis and therapy.

1. The study provided comprehensive analysis of aortic stiffening at different time points and sex groups. The temporal correlation between Lox induction and ECM remodeling associated with aortic physiology are well performed and supported. However, the authors only tested the expression in aortic tissue, but showed Lox inhibition conferred significant improvement in cardiac stiffness as well. Yet, Lox expression and activity in cardiac tissue, along with histological measurement were not provided. Other markers of cardiac dysfunction, including marker genes should be analyzed to support the functional improvement in treated mouse hearts. Adding these data would strengthen the conclusions and impact of the finding.

2. miR145 was identified as a conserved regulator for LOX through correlated gene expression profile and gain of function analysis in culture. This is an important finding of this report, and the conclusion should be better substantiated by a loss of function analysis using antagomer for miR145 and the utilize a reporter assay to support the direct regulatory mechanism.

3. Authors need to provide better description of the RNA analysis, how many samples were sequenced, how reproducible of each sample in global gene expression patterns (illustrated by PCA) and how many total DEG transcripts were identified. What is the outcome of IPA analysis in terms of cellular processes and pathways affected in the HGPS tissues. In short, more comprehensive transcriptome data analysis would add much value and impact to this report as well.

Richard K. Assoian, Ph.D.

Professor
Director, Program in Translational Biomechanics
Institute of Translational Medicine and Therapeutics
co-Director, NSF Science and Technology Center for
Engineering Mechanobiology

Dr. Shachi Bhatt, Executive Editor

Life Science Alliance

Dear Dr. Bhatt:

Thank you for your positive comments and those of the arbitrator. We were delighted to hear that you look forward to publishing our paper in Life Science Alliance.

We have followed your advice and have added data requested by the arbitrator when available or toned down the respective conclusions and addressed the limitations in the Discussion. We have provided a better description of the RNA analysis and submitted the primary data to a public repository as per your request. All of these points are specifically addressed below.

Arbitrator Comments

1. Lox expression and activity in cardiac tissue, along with histological measurement were not provided. Other markers of cardiac dysfunction, including marker genes should be analyzed to support the functional improvement in treated mouse hearts.

REPLY: The COVID-19 pandemic has significantly limited our work on the heart. As this data is not readily available, we toned down our conclusions and stated limitations and the need for additional work in the Discussion (lines 343-348).

2. miR145 was identified as a conserved regulator for LOX through correlated gene expression profile and gain of function analysis in culture. This is an important finding of this report, and the conclusion should be better substantiated by a loss of function analysis using antagomer for miR145 and the utilize a reporter assay to support the direct regulatory mechanism.

REPLY: In fact, we previously showed that a miR-145 antagomer regulates LOX mRNA levels in smooth muscle cells and that the direction of its effect is complimentary to that of ectopic miR-145 expression. But we thank the arbitrator for making us realize that those prior results were not stated in this manuscript. We now describe and cite that prior study with LOX and the miR145 antagomer (Discussion: lines 359-362).

We do not have available data from a reporter assay, so have modified the Discussion to indicate that such a result could further strengthen the connection between miR-45 and LOX mRNA abundance (Discussion; lines 365-367).

3. Authors need to provide better description of the RNA analysis, how many samples were sequenced, how reproducible of each sample in global gene expression patterns (illustrated by PCA) and how many total DEG transcripts were identified. What is the outcome of IPA analysis in terms of cellular processes and pathways affected in the HGPS tissues. In short, more comprehensive transcriptome data analysis would add much value and impact to this report as well.

REPLY: As requested, we have provided a better description of the RNA analysis. We added the requested information about sample size to Material and Methods (line 488) and the approximate number of DEGs to Results (line 253). We have also provided the requested PCA and pathway (gene ontology) analysis (new Figure S13A-B, Results lines 254-256, and Materials and Methods lines 500-502 and 503-504). Finally, we have deposited the RNASeq files into GEO and provide the accession number (lines 500 and 535).

We hope the new data, the more accurate description of our previous work with a miR-145 antagomer, and the modifications we have made to the text make our study acceptable for publication in Life Science Alliance.

Sincerely,

Richard K. Assoian, Ph.D

February 23, 2021

RE: Life Science Alliance Manuscript #LSA-2020-00997-TR

Author information redacted

Dear Dr. Assoian,

Thank you for submitting your Research Article entitled "Arterial stiffness in Hutchinson-Gilford Progeria Syndrome corrected by inhibition of Lysyl Oxidase". It is a pleasure to let you know that your manuscript is now accepted for publication in Life Science Alliance. Congratulations on this interesting work.

DISTRIBUTION OF MATERIALS:

Again, congratulations on a very nice paper. I hope you found the review process to be constructive and are pleased with how the manuscript was handled editorially. We look forward to future exciting submissions from your lab.

Sincerely,

Shachi Bhatt, Ph.D.
Executive Editor
Life Science Alliance

<https://www.lsjournal.org/>

Interested in an editorial career? EMBO Solutions is hiring a Scientific Editor to join the international Life Science Alliance team. Find out more here -

https://www.embo.org/documents/jobs/Vacancy_Notice_Scientific_editor_LSA.pdf